# Detecting Machining Defects inside Engine Piston Chamber with Computer Vision and Machine Learning

**DOI:** 10.3390/s23020785

**Published:** 2023-01-10

**Authors:** Marian Marcel Abagiu, Dorian Cojocaru, Florin Manta, Alexandru Mariniuc

**Affiliations:** Faculty of Automation, Computers and Electronics, University of Craiova, 200585 Craiova, Romania

**Keywords:** computer vision, sensors, machine learning, industry, manufacturing, robotics

## Abstract

This paper describes the implementation of a solution for detecting the machining defects from an engine block, in the piston chamber. The solution was developed for an automotive manufacturer and the main goal of the implementation is the replacement of the visual inspection performed by a human operator with a computer vision application. We started by exploring different machine vision applications used in the manufacturing environment for several types of operations, and how machine learning is being used in robotic industrial applications. The solution implementation is re-using hardware that is already available at the manufacturing plant and decommissioned from another system. The re-used components are the cameras, the IO (Input/Output) Ethernet module, sensors, cables, and other accessories. The hardware will be used in the acquisition of the images, and for processing, a new system will be implemented with a human–machine interface, user controls, and communication with the main production line. Main results and conclusions highlight the efficiency of the CCD (charged-coupled device) sensors in the manufacturing environment and the robustness of the machine learning algorithms (convolutional neural networks) implemented in computer vision applications (thresholding and regions of interest).

## 1. Introduction

Computer vision applications are being used intensively in the public area for tedious tasks, e.g., surveillance and license plate detection and reading, as well as in robotics applications for tasks, e.g., object detection, quality inspection, and human machine cooperation [1,2,3].

In the initial stages of development, implementing a computer vision application (machine vision or robotic vision versions) was considered an exceedingly challenging task. With the increase of the processing power, new hardware development, and new, efficient, and performant image sensors, the development of such applications was made significantly easier [4,5].

A huge boost in popularity for the image processing and computer vision application was achieved with the increase in popularity of Python programming language and the implementation of various image processing frameworks such as OpenCV (for C++ initially and Python afterwards) and the development of machine learning and deep learning frameworks [6,7].

Solutions implemented in the robotic manufacturing environment are based on cameras using CCD sensors and industrial systems, which consider the computer vision application as a black box providing a status. This approach proved to be robust and efficient. The needs of industry are now growing different and becoming more complex. The control systems also need to integrate with computer vision applications to provide full control for the production process [8,9].

The current global and geopolitical context from the last years, the tendency for accelerated car electrification, and recent innovation from Industry 4.0 have encouraged car manufacturers to integrate more computer vision applications in the production process. Applications are mostly used for counting parts and ensuring traceability, e.g., barcode readings, QR code readings, OCR, and defect detection in distinct stages of the manufacturing process, e.g., painting, assembly, and machining. In this environment more complex applications can be found, e.g., high precision measurement tools based on computer vision, complex scanning, and applications based on artificial intelligence (machine learning) [10].

The solution presented in this paper is based on the integration of a CCD sensor camera with a robotic control system that is also able to provide all the information needed in the robotic manufacturing environment for traceability and planning while detecting complex defects in real time. Two algorithms are used for detecting a class of defects inside the cylinder chamber of an engine block. The main role of the computer vision algorithms is the reducing the number of input features for the convolutional neural network by isolating the region of interest (walls of the cylinder chamber). The convolutional neural network scope is to process the newly generated image for providing a decision.

The future actions of the entire robotic system that manipulates these mechanical parts depends on the results provided by the visual inspection system. Moreover, based on the global results obtained on the entire visual inspection process, reprogramming or even reconfiguration of the robotic systems involved in the manufacturing process of mechanical parts will take place [11].

In order to implement this solution, the goal was to develop a computer vison system that is able to detect machining defects from the cylinder chamber of the engine block. This was achieved by developing the following steps:The system with reused parts.A new system architecture based on the available parts.A new software architecture to match the industrial standards.A new user interface for the software application.Integrating and updating the software development kits of the camera and input/output ethernet module.An algorithm to isolate the region of interest in the acquired image.A machine learning algorithm able to receive an input in the format generated by the previous computer vision algorithm.

## 2. Related Work

Defect detection technologies are used in the manufacturing industry for identifying the surfaces (spots, pits, scratches, and color differences) and internal parts (defects holes, cracks, and other flaws) of the products having problems. Computer vision defect detection applications must be fast, non-destructive, and accurate, and they have become widely used in the recent years. Zhou et al. [12] developed an artificial imaging system for the detection of discrete surface defects on vehicle bodies using a three-level scale detection method for extracting the defects that might appear on the vehicle surface. The method distinguishes the defect location, comparing the features of the background from the defect images, which allows for detection in concave areas or areas with abrupt changes in the surface style lines, edges, and corners. It extracts defects that are hardly perceived by the human eyes.

In various computer vision industrial applications, the basic setups for image acquisition are similar. For example, in the automotive manufacturing industry, a basic computer vision application is needed a light source alongside a camera and a computer powerful enough to process the acquired image. As light sources, LEDs are mostly used. LED light sources offer high efficiency and versatility when it comes to triggers and dimming control. Infrared light sources used with monochrome industrial cameras (or as infrared panels) as well as multiple light sources are frequently used. For settings and environment closer to the laboratory, in the majority of the computer vision application, cameras and light sources are placed in a light absorbing room where the lighting can be controlled. A special application, e.g., an assembly robot, may require a special camera. In this case, the light source and the camera will be attached to an actuator (servomotor, robotic arm, etc.). Industrial cameras contain CCD (charge-coupled device) or CMOS (complementary metal-oxide semiconductor) sensors and the lenses are chosen having in focus the environment and the vision application. Trying to achieve real time processing, the software algorithms must be executed on powerful machines. Algorithms are developed by customizing to each particular application and each hardware configuration (camera and lighting). For detecting different defects of a car after the painting process, a four-camera setup can be used to achieve stable light and multiple cameras (e.g., five cameras) to acquire the same affected area from multiple angles (light conditions different). In the acquired images, the region on interest will be isolated, several specific filters for noise reduction will be also applied, in addition to a feature extraction algorithm (specific to the vision application) for isolating the different defects detected [11].

When a certain amount of data can be acquired and used, then a deep learning model training supervised learning is adopted instead of a conventional recognition based on feature descriptors. A classification module, an integrated attention module with an image segmentation module, is used for weekly supervised learning. The classification module extracts the defect features from the image. The integrated module has as a purpose the detection of different irregular defects (e.g., for metal pieces) which can appear after casting or shaping processes. The segmentation module is used to determine if a pixel from the image is associated to a defect area [13].

Other common defect detection methods are ultrasonic testing, osmosis testing, and X-ray testing [14]. The ultrasonic methods are used in the detection of defects in the internal structure of the product under test (like X-ray testing). These methods are based on filtering for feature extraction and the ability to describe the identified defect.

Alongside common methods, in recent years, deep-learning defect detection methods have been used in various applications. Some of these algorithms are based on the use of a deep neural network, e.g., a convolutional neural network, residual networks, or recurrent neural networks. Computer vision defect detection applications have shown good accuracy in binary defect detection [15].

In their paper, Zhonghe et al. [16] address the state of the art in defect detection-based machine vision, presenting an effective method to reduce the adverse impact of product defects. They claim that artificial visual inspection is limited in the field of applications with possible dangerous consequences in the case of a failure because of the low sampling rate, slow real-time performance, and the relatively poor detection confidence.

The replacement of artificial visual inspection is machine vision, which can cover the whole electromagnetic spectrum, from gamma rays to radio waves. Machine vision has a great ability to work in harsh environments for a long time and greatly improves the real time control and response. Therefore, it can improve many robotic manufacturing processes to support industrial activities. In this paper, the proposal for an industrial visual inspection module consists of three modules: optical illumination, image acquisition, and the image processing and defect detection module. It is stated that an optical illumination platform should be designed. Then, CCD cameras or other acquisition hardware should be use in such a way that the information carried by them to the computer should have an extremely high quality. Finally, either classical image processing algorithms or better, deep learning algorithms should be used, which are able to extract the features and perform the classification, localization, segmentation, and other image operations, image processing being the key technology in machine vision. In industry, this architecture can be used as a guideline for designing a visual inspection system. It is given as an example in the paper for inspecting surface characteristics in designing a highly reflective metal surface.

Wang Liqun et al. [17] focused on the idea of detecting defects using deep learning. They also based their research on convolutional neural networks for training and learning big sets of image acquisition data and they claim that it can effectively extract the features and classify them accurately and efficiently. They use PatMax software, which can recognize twenty-five different filter shapes, and determines the location of the filter while being 99% accurate. The process first collects the information from the camera, reads the image preprocessed result and trains on the processed images, then establishes a complete information model and obtains the target image. A diffuse bright led backlight illumination is used. Light sensitive components were used for image acquisition, and a wavelet smoothing was used for image preprocessing, after which Otsu threshold was used to segment the image. In the end, the support vector machine classifier was designed for defect classification. The goals should be high precision, high efficiency, and strong robustness. Therefore, the system needs an excellent coordination of the three modules. The features are afterwards matched with the template, and the quality of the assembly process is judge according to the matching result. Difficulties remain in detecting component defects due to the variety of vehicle parts which have different shapes, and due to the fact, the defects are very diversified. Moreover, the image structure of the parts is more complex, incorporating irrelevant factors around the image and a lot of noise, which makes feature extraction difficult. The authors managed to improve the VGG16 network model structure by adding the inceptionv3 module, increasing the width of the model based on depth. Their resulted accuracy was improved from 94.36% up to 95.29%, which is almost 1% more accurate than previously.

Zehelein et al. [18] presented in their paper a way of inspecting the suspension dampers on the autonomous driving vehicles between inspections. Their theory claims that in a normal vehicle, the driver always monitors the health state and reliability of a vehicle, and that it could be dangerous for an autonomous driving vehicle to not be monitored between inspections. To solve this problem, they discussed one of the problems in defect diagnosis while driving, namely the robustness of such a system concerning the vehicle’s configuration. The main problems are the variable factors, such as tire characteristics, mass variation, or varying road conditions. They decided to combine a data driven approach with a signal-based approach, which led to a machine learning algorithm which can incorporate the variations in different vehicle configurations and usage scenarios. In their paper, it is stated that they used a support vector machine for classifying signal features, and they also needed features that can distinguish between different health states of the vehicle. Convolutional neural networks can deal with multidimensional data and demonstrate good feature extraction, which makes them perfect for the job. They used the driving data of the longitudinal and lateral acceleration as well as the yaw rate and wheel speeds. Using FFT (fast Fourier transform), input data were shown to give the best results regarding classification performance.

The authors were not able to check the real time implementation of the system because there is not a specific value for the computing power of an automotive ECU (electronic control unit). Therefore, the algorithm might not run optimally for every vehicle on the market [19]. They also propose the feature extraction method and divide the defects into three categories (pseudo-defects, dents, and scratches) using the linear SVM (scan velocity modulation) classifier. Their detection results were close to the accuracy of 95.6% for dents and 97.1% for scratches, while maintaining a speed of detection of 1 min and 50 s per vehicle. They state that their system could be improved using deflectometry techniques for image defect contrast enhancement, or by improving the intelligence of the method, but the latter could slow down the detection speed. Moreover, if they could use parallel computing techniques on graphic processing units, the speed of detection could be further improved.

A conventional computer vision approach is implementing the following algorithm [20,21]:Image acquisitionCircle and point edge detectionLength and radius measurementsFeature collectionMatching featuresGenerate the verdict and store it in a database

A drawback of this approach would be the high processing time of the high-resolution input image and the volatile environment from where the image is acquired, which will lead to a repeatability issue in the image due to dynamic shapes and contrast of the emulsion marks [22].

In this paper, a description of a combination between a conventional approach and a machine learning approach is given.

## 3. Solution Overview

### 3.1. Process and Issue Description

The purpose of the inspection machine is to detect a certain class of defects, to sort the engine blocks on the production line, and to wash the bottom part of the block using a special system for removing dirt, dust, and other mechanical process impurities. When a defect is detected, the engine block is removed trough a conveyor from the production line. In the washing process, some special solvents and emulsions are used.

The CCD sensor camera was configured to match a high range of environment conditions with a fixed exposure, fixed gain, and a special set of lenses. The specification of the lens used are further described in Figure 1. The implementation completes the already installed inspection machine by adding a new station with the purpose to automate the visual inspection performed until now by the operator. The complete layout of the process can be observed in Figure 2.

The next step after washing will be the drying and cleaning of the block with an airflow through the bottom part and the cylinder chambers. The drying process leaves behind dried cleaning emulsion, which will make the automated inspection more difficult. In Figure 3a,b, the traces of dried emulsion can be observed on a flawless cylinder chamber. Figure 4a,b describes the defect to be automatically detected from the cylinder chamber alongside dried emulsion.

The engine block is made from cast iron with green sand insertions. In the process of filling the mold with liquid metal, some environment factors can interact with the product in an unwanted way. The damaged sand core can generate inclusions inside or at the surface of the part. Another defect is generated by the impossibility of removing all the gases from the mold when the liquid metal takes contact with the surface of the mold. This process involves generating blowholes.

### 3.2. Architecture Description

Figure 5 describes the system architecture including the sensor. The solution was implemented using a single camera capturing an image of the area that needs to be inspected. For moving the camera, a PLC that is controlled directly by the main inspection machine was used. When the PLC receives the trigger, a stepper motor is actuated. The camera is hovered over each of the cylinders for acquisition and is connected to the stepper motor with a coupling in order to create a linear movement. When the camera is in position, the acquisition and processing system is triggered.

For ensuring a more increased degree of repeatability in the image acquisition, an infrared LED flash is used. The LED is controlled by the acquisition system. The main controller used for acquisition and processing is represented by a Jetson Nano development kit, which has a high computing power, especially in artificial intelligence applications. The Jetson interacts with the PLC trough an industrial remote I/O module from Brainbox by controlling it over ethernet. The Brainbox module also triggers the LED flash. The control of the CCD camera is also implemented over Ethernet, in this case POE (power over Ethernet) because the camera is also powered by the ethernet switch.

### 3.3. Hardware Description

For implementing the computer vision solution, the following hardware components were used:Nvidia Jetson Nano controller with 4 GB RAM memory, 128 core GPU, an ARM Cortex A57 Quad-Core CPUThe Imaging Source DMK 33GX290e Monochrome CameraED-008 Ethernet to Digital I/O Brainboxes moduleEffiLux LED Infrared Light flash EFFI-FDIndustrial compliant POE Ethernet Switch

The Jetson controller is connected to the Ethernet switch alongside the camera with a CCD sensor and remote I/O module. The LED Flash is connected with a pull-up resistor (24 V) to the remote I/O module and is controlled via ethernet requests by the Jetson controller.

### 3.4. Software Description

The application was implemented using Python programming language. All the used hardware components integrate Python software development kits provided by the manufacturer. Therefore, the choice of programming language for implementation came naturally. The human–machine interface was implemented using the PyQt framework (PyQt5-Qt5 version 5.15.2, developed by Riverbank Computing, open-source), which is a Python wrapper of the open-source framework Qt developed by the Qt company. Software was designed to cover all of the manufacturing necessities, e.g., logging, user management, process handling, and so on [21,22,23,24,25,26,27,28,29,30]. In Figure 6, a complete sequence diagram of the process can be observed.

### 3.5. Processing Algorithm Description

The processing algorithm has two main parts: the conventional processing and the prediction using a machine learning model. As input, the algorithm takes a grayscale image with a resolution of 1920 × 1200. In the conventional processing, the ROI of the inner chamber of the cylinder is extracted by the algorithm, normalized, and a gaussian filter is applied. After applying the filter, an adaptive thresholding is also performed by the algorithm because the defects have a lower grayscale level and can be isolated this way. When the defects are isolated by the thresholding, they are marked with a contour function. This function returns the area of the detected contours (each contour detected represents a possible defect).

The area can be evaluated for establishing a verdict. The conventional processing works verry well when there are no significant traces of cleaning emulsion on the cylinder. When the emulsion becomes mixed with dust, traces become increasingly noticeable and with a lower grayscale level. Because of that, the thresholding is no longer able to distinguish between traces of emulsion and actual defects [22,23,24,25,26,27,28,29,30].

The second part of the processing algorithm is the convolutional neural network implemented using the PyTorch framework. The first layer takes as input the three RGB channels of the image and splits it in eight features for the next layer. The second layer is a max pooling layer with a kernel size of 3 × 3 and with padding enabled for looping through the entire image. Third layer is another convolutional layer, similar to the first layer, followed by a fully connected layer [31,32,33].

The spatial size of the output is obtained:(1)$$\frac{W-K+2P}{S}$$
where *W* is the volume input size, *K* is the kernel size of the convolutional layer neurons, *S* represents the stride, and *P* is the amount of zero padding at the borders of the neural network [20].

A typical pooling layer is calculated with the following formula:(2)$$f\left(x,y\right)=\max_{a,b=0}^{1}\;S_{2x+a,\;2y+b}$$

The activation functions for the convolutional layers are ReLu (rectified linear unit), applying the following non-saturating activation function for removing the negative values from the activation map by setting them to zero [20].
(3)$$f\left(x\right)=\max\left(0,x\right)$$

In Figure 7, an architecture of the neural network is proposed [19,20,21,22,23,24,25,26,27,28,29,30].

Hyper-parameters:Input channels—3Convolutional layers—5Fully connected layer—1Batch size—8Epochs—30Output channels—2

## 4. Results

The model was trained using old defective engine blocks as well as on fixed periods of time with new batches of images evaluated by the model as defects. The false defects were labeled as no defects in the dataset and the actual defects were added in the dataset. The model did not perform very well, as can be observed in the results, due to the high number of features that needed to be extracted and processed before setting a verdict.

The architecture presented in Figure 7 has five convolutional layers with max pooling and a fully connected layer. The end image now has a resolution of 60 × 37 with 128 unique features. In Figure 8, we can see that the loss function generated during training with the new model has better performance and is able to detect the defects much faster.

Below, the training results can be observed:Accuracy on training set: 100%Accuracy on test set: 100%Loss at the end of the training: 0.13

The main indicators tracked during commissioning was the number of the false detections reported by the neural network and the rate of detection for real defects. The number of false detections was initially high due to emulsion marks and system calibrations (refer to Figure 9). After the dataset was established and the system calibrated, the indicator decreased substantially, below a desired threshold such that we can consider that the algorithm is reliable enough.

It was observed that after including images with prominent marks of emulsion and with small imperfections generated by improper lighting (camera stabilization), the number of false detections decreased considerably (refer to Figure 10 and Figure 11).

Table 1 also shows the type of defects detected during operation in relation to false detections and real defects. It can be observed that the highlight is still on the emulsion marks resulting from the washing and drying process in the presence of the dust or other impurities, this being the main false detection generator.

There is a lot of research being conducted on how different algorithms are responding to already established datasets. The main approach used involves software pre-processing, the use of a convolutional neural network for feature extraction, and in some cases another network for classification [34,35,36,37,38,39,40,41,42]. 

This solution is uses only hardware pre-processing (camera as sensors and environment related) and one convolutional neural network for feature extraction and classification. The setup proved to be sufficient and robust for the needed classification.

## 5. Conclusions

From the point of view of robotics applications developed in the automotive industry, the robustness of image processing applications from the manufacturing area can be increased considerably by using a machine learning algorithm to replace the classic method of processing with filters, hardware, complicated optics, and complex software algorithms. The machine learning algorithm can replace the classic approach and thereby ensure greater flexibility in developing the backbone of the application, e.g., PLC communication, socket services, and human–machine interface, so indispensable in this environment.

The weak point of this implementation remains the dependency on a sufficient and a correct dataset. By ensuring that we have the correct data to work with, we can develop and train a robust application for use in the manufacturing environment.

The advantage of using such an approach is that other implementations, e.g., communication, HMI, logging, and others, can be abstracted and reused in new implementations. The training of the algorithm can also be abstracted and reused. The flexible part needs to remain the architecture of the neural network and the dataset used.

Based on the work presented in this paper, a new application is already in development. The scope of the new implementation is to detect the presence of a safety pin inside the piston locking mechanism. This operation will be performed using three cameras which are triggered simultaneously for acquisition, a new architecture for the neural network, and different hardware to support a more complex application.

## Figures and Tables

**Figure 1 sensors-23-00785-f001:**
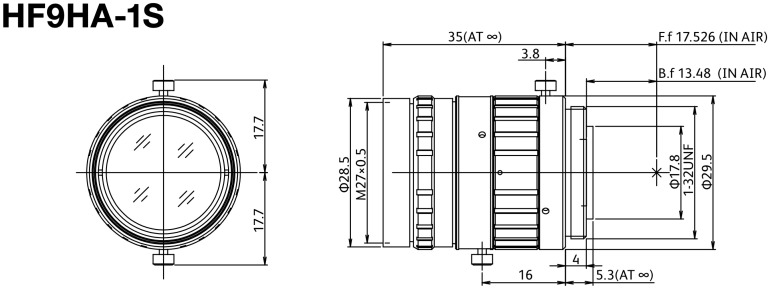
Fujinon lens dimensions.

**Figure 2 sensors-23-00785-f002:**
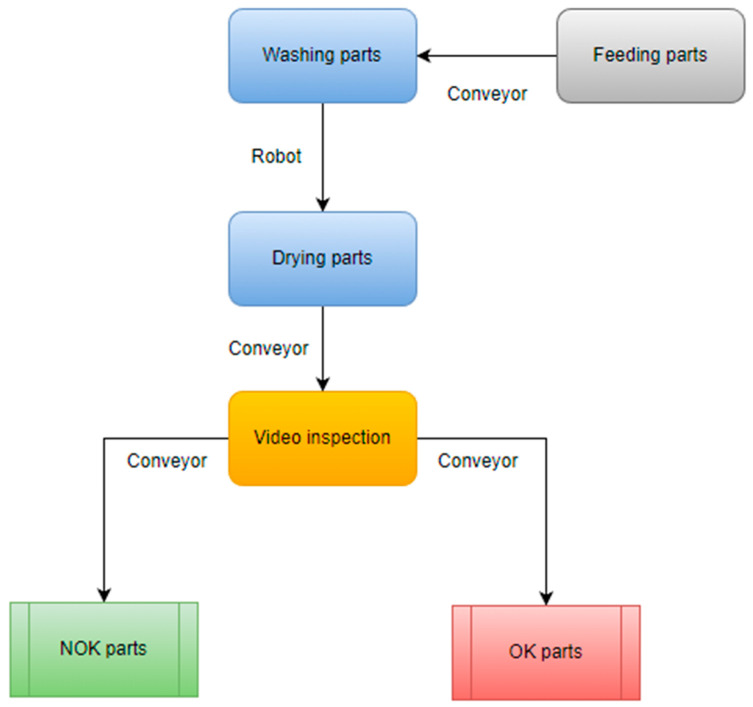
Architecture of the washing and sorting machine on the production line.

**Figure 3 sensors-23-00785-f003:**
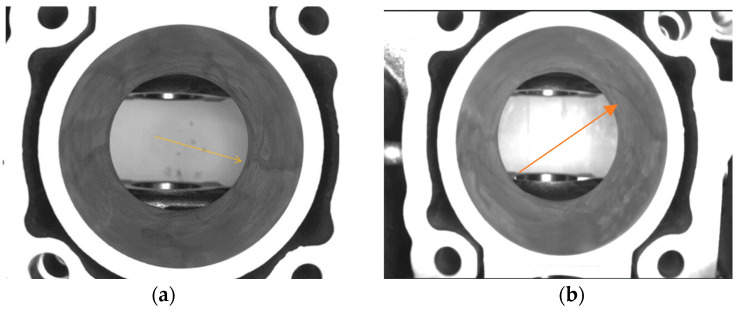
No defects in the cylinder chamber. In (**a**,**b**) can be observed a part with no defects and dried emulsion marks indicated by the arrows.

**Figure 4 sensors-23-00785-f004:**
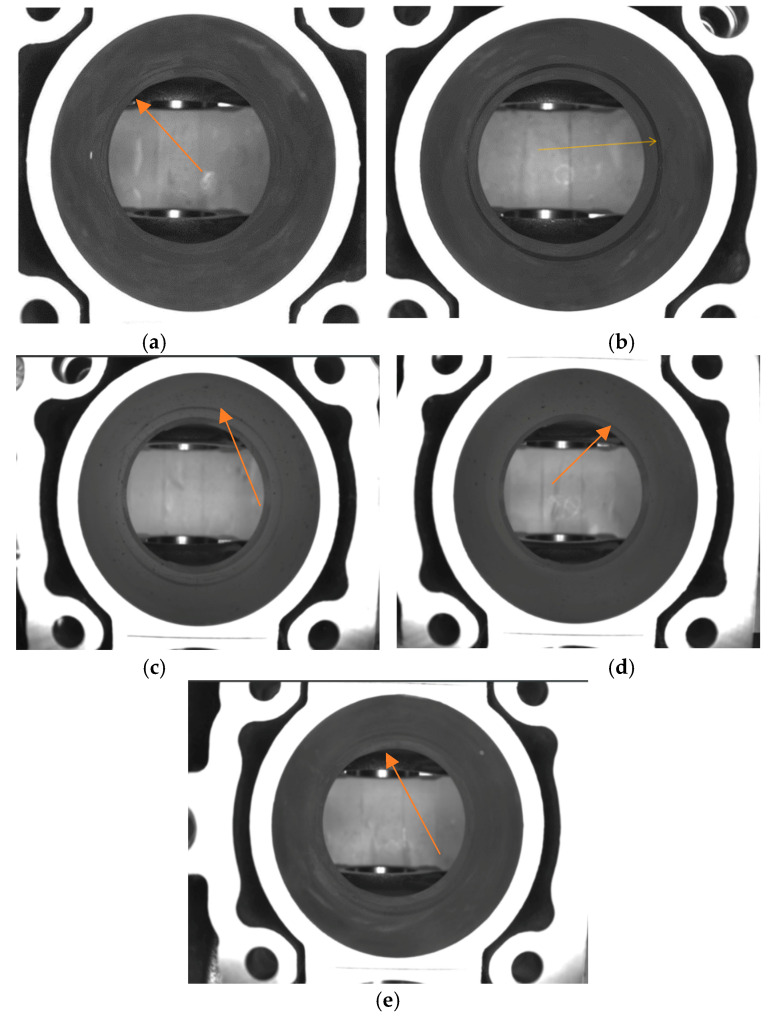
Defects in the cylinder chamber. In (**a**–**e**) parts with machining defects can be observed. (**a**) presents a barely observable defect and (**b**–**e**) presents a more prominent one.

**Figure 5 sensors-23-00785-f005:**
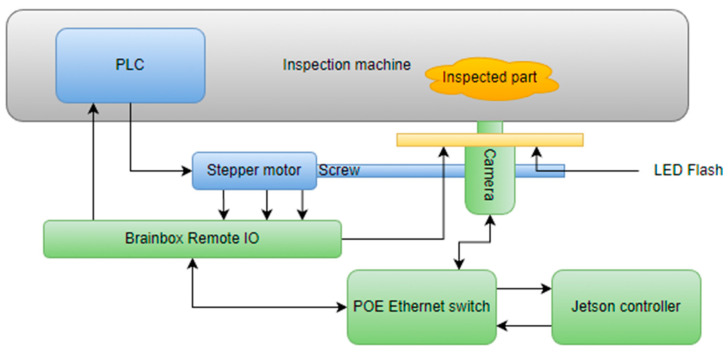
System architecture of the video inspection operation.

**Figure 6 sensors-23-00785-f006:**
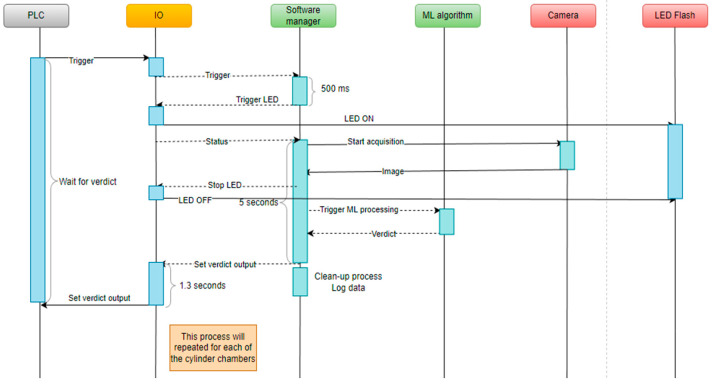
Process sequence diagram.

**Figure 7 sensors-23-00785-f007:**
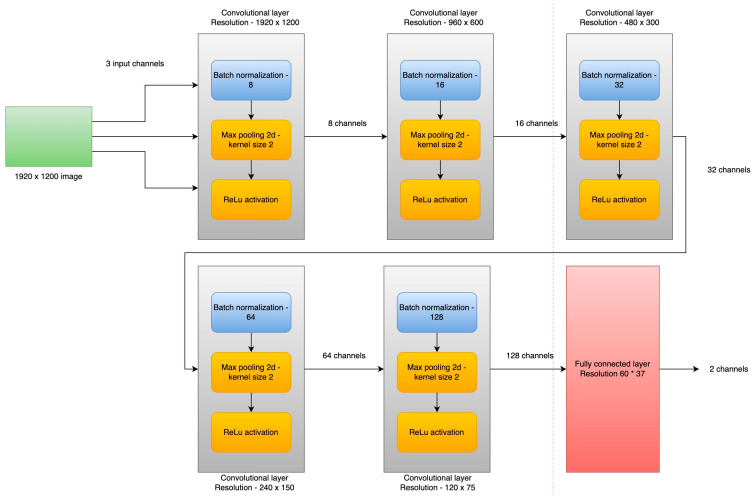
Convolutional network architecture.

**Figure 8 sensors-23-00785-f008:**
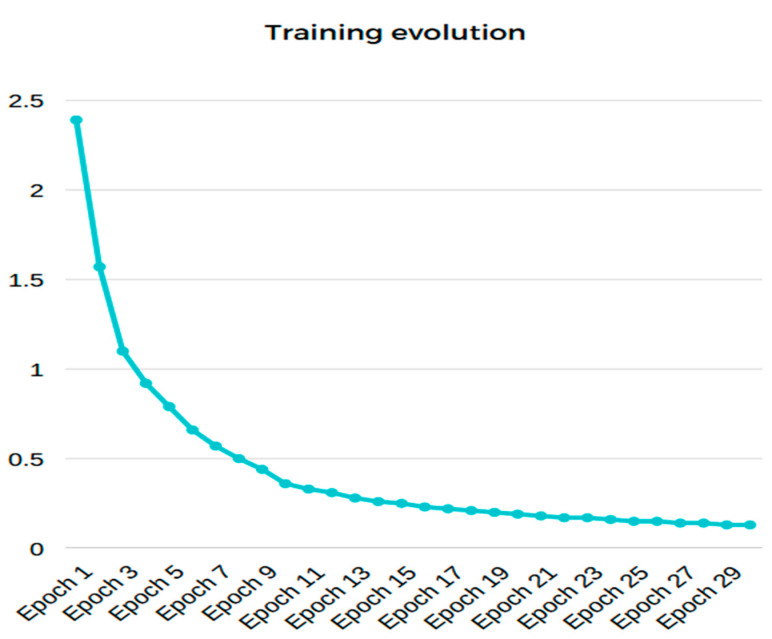
Training progress of the convolutional neural network. The X-axis describes the loss function value can be observed. The Y-axis shows the corresponding epoch.

**Figure 9 sensors-23-00785-f009:**
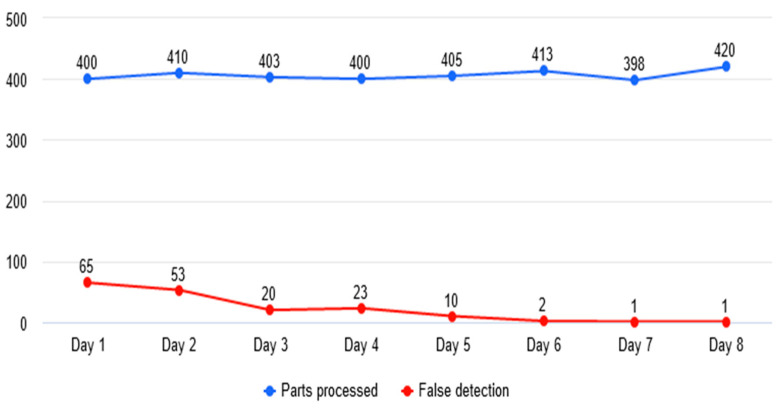
Evolution of false detections.

**Figure 10 sensors-23-00785-f010:**
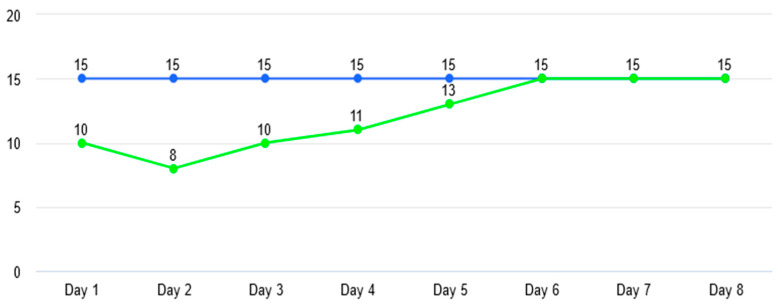
Evolution of detections. Blue line represents the real number of defects provided to algorithm and the green line represents the defects detected by the algorithm over 8 days.

**Figure 11 sensors-23-00785-f011:**
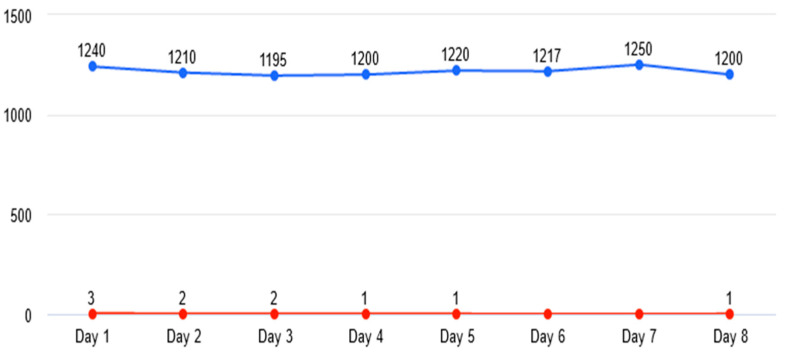
False detection evolution after final training.

**Table 1 sensors-23-00785-t001:** This table shows the evolution of detections.

Number of Parts	False Detections	Actual Defects	Error Type	Defect Type	Remarks
1200	100	10	Emulsion marks	Machining defect	Tests performed
1160	50	0	Emulsion marks	N/A	N/A
1193	52	0	Emulsion marks	N/A	N/A
1203	37	0	Emulsion marks and lighting issues	N/A	N/A
1210	10	0	Highly intense emulsion marks	N/A	N/A
1205	1	1	Highly intense emulsion marks	Machining defect	Tests performed

## Data Availability

Our dataset is not publicly available.

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
