# Peer review of "Detecting Machining Defects inside Engine Piston Chamber with Computer Vision and Machine Learning"

_sensors, 2023, doi:10.3390/s23020785_

Round 1
Reviewer 1 Report
This paper proposes a method to detect the machining defects from an engine block using computer vision and machine learning. However, the methods are not adequately described, the results are not clear and there are many typo errors. The whole paper seems like a technique report instead of academic paper. There are some advices to improve the quality of this paper.
Q1. In line 213, the author said “In Figure 2 it can be observed the traces of dried emulsion on a flawless cylinder chamber.”
It seems that Fig. 2 does not offer any information of flawless chamber or dried emulsion.
Q2. In Fig. 2, how to distinguish the OK parts from the NOK parts?
Q3. Actually, I cannot tell any machining defect in Fig.3 (a) and emulsion marks in Fig. 3(b) according to the author's arrow. Plus, what is the difference between Fig. 3 and Fig. 4?
Q4. The flow chat in Fig.6 is confusing. It has a start but it does not show an end. Could the author make it clearer?
Q5. There is only one curve in Fig. 8, and the author did not point out the label of X-axis and Y-axis, how could he conclude that "the new model has better performance"?
Q6. In the reference part, there are two references in Reference 1.
Author Response
Dear All,
The authors of this paper respectfully thank the evaluators for the observations made in order to increase the quality of the work and manage to pass over the threshold imposed for publication.
The authors did their best to put into practice the observations received.
A brief overview of the changes made is presented in the attached file.
Thank for your help,
Prof. Dr. Eng. Dorian Cojocaru

Reviewer 2 Report
After reading the paper, the reviewer has the following comments:
1) Abstract: Clearly state the used computer vision (CV) and machine learning (ML) methods. State the performance of the proposed approaches in terms of prediction accuracy. Avoid using abbreviations without mentioning their full technical terms.
2) Abstract: Revise the abstract as a single paragraph.
3) Regarding “Computer vision applications are being used intensively in the public area for tedious tasks like surveillance, license plate detection and reading, and as well in the robotics applications for tasks like object detection, quality inspection, human machine cooperation.”, use several references to support the point of discussion.
4) Regarding “In the initial stages of development, implementing a computer vision application (machine vision or robotic vision versions) was considered an exceedingly challenging task; with the increase of the processing power, new hardware development and new efficient and performance image sensors, the development of such applications was significantly made easier.”, use several references to support the point of discussion. In addition, please correct the typos: “performant image sensors”.
5) Regarding “A huge boost in popularity for the image processing and computer vision application was achieved with the increase in popularity of Python programming language…”, use several references to support the point of discussion.
6) Regarding “ “Solutions implemented in the robotic manufacturing environment are based on cameras 44 using CCD sensors and industrial systems, which are considering the computer vision 45 application as a black box providing a status. This approach proved to be robust and 46 efficient.”, use several references to support the point of discussion.
7) Regarding “like high precision measurement tools based on com- 56 puter vision, complex scanning and applications”, please revise the typo as follows: “…based on artificial intelligence (machine 57 learning) [1] .”
8) At the end of the section 1, please add a short paragraph to discuss the used CV and ML methods as well as their roles in the proposed framework.
9) At the end of the section 1, please add several bullet points to summarize the contributions of the paper.
10) Regarding “Thomas Zehelein, Thomas Hemmert-Pottmann and Markus Lienkamp presented in ….”, provide the number of the reference of interest. Please check the similar problem throughout the paper, such as “Qinbang Zhou, Renwen Chen, Bin Huang, Chuan Liu, Jie Yu and Xiaoqing Yu …”. It is recommended that “Zehelein et al.” or “Zhou et al.” are used instead of the full lists of authors.
11) In section 2, the authors have reviewed a number of related works. However, the literature review section is still not comprehensive. Please review more papers and discuss about the use of CV and ML in surveying/assessing the condition of mechanical systems which are similar to the investigated problem of detecting machining defects inside engine piston chamber. The review should focuses on the works related to assessing engine piston chamber or similar components. Review of conventional approaches and their limitations are welcome. Based on the literature review, the drawbacks of existing works should be summarized and the authors should point out the gaps in the literature that should be addressed in the current work. A table that summarizes the existing works is very useful for the readers.
12) In section 1 or section 2, please summarize the commonly found defects in engine piston chamber.
13) In section 1 or section 2, what are the defects that the proposed system trying to recognize? Please briefly mention this detail.
14) In 3.1. Process and issue description, if possible, please provide the images to illustrate each step of the process of interest. These images can help reader better capture the setup of the experiment.
15) Fig. 6 needs more explanations:
(a) Please add more details of the acquired images (format, number, etc.). Please provide more examples of these images.
(b) What are the trained models?
(c) What are the equations used to compute the inference score?
16) Fig. 7: “Figure 7. First Convolutional network architecture” should be “Figure 7. The used convolutional neural network architecture”
17) Describe the method for setting the hyper-parameters (e.g. the number of the convolutional-pooling layers) of the convolutional neural network?
18) Was batch normalization used?
19) Why did the author attempt to train a new convolutional neural network instead of using deep transfer learning?
20) “as ca be observed in the results chapter” should be “as can be observed in the results chapter”
21) Please revise the Figure 8. Loss function: add the unit of the Y-axis and X-axis. The values along the X-Axis look weird as they should be integer. The title of this figure should be the training progress of the convolutional neural network.
22) Is it possible to reduce the number of false detection by training the neural network to recognize the emulsion marks as well as lighting issue?
23) Provide a better assessment of the system performance by the use of indices such as precision, recall, F1 score, and Cohen’s Kappa coefficient.
24) Please proofread the whole paper to improve the grammar and correct all the typos.
Author Response

(The authors gave the same response as above.)

Round 2
Reviewer 1 Report
There are still some typo and grammatical mistakes, e.g. In line 72, "This was task is supported by developing the following steps." Please proofread to improve the quality of this paper.
Author Response
Dear All,
The authors of this paper respectfully thank the evaluators for the observations made in order to increase the quality of the work and manage to pass over the threshold imposed for publication.
The authors did their best to put into practice the observations received.
A brief review of the changes made for minor revisions is presented below.
Thank for your help,
Prof. Dr. Eng. Dorian Cojocaru

Reviewer 2 Report
The reviewer appreciates the author's effort in revising the paper.
There are several minor comments:
1) On page 11, line 360: Please remove the discussion regarding the transfer learning ("... and a transfer learning to a new architecture was not feasible."). In fact, transfer learning is always feasible for a image recognition task. For instance, we can apply squeezeNet or AlexNet to recognize the defects of interest. If deep transfer learning is not within the scope of the current work then it is fine to skip the discussion regarding this method. However, it is recommended that the authors consider its use in a future work.
2) On page 11 line 370-371: Consider revising the grammar of the sentences. They should be "The x-axis describes ... The y-axis shows ..."
3) Line 384: "decreased considerably (refer to Figure 11)."
4) At the end of the section 5, consider adding some discussions regarding the future extensions of the current work.
Author Response

(The authors gave the same response as above.)
